



# The wave spectrum in archipelagos

Jan-Victor Björkqvist[1], Heidi Pettersson[1], and Kimmo K. Kahma[1]

[1]Finnish Meteorological Institute, P.O. Box 503, FI-00101, Helsinki, Finland

*Correspondence to:* Jan-Victor Björkqvist (jan-victor.bjorkqvist@fmi.fi)

**Abstract.** Sea surface waves are important for marine safety and coastal constructions, but mapping the wave properties at complex shorelines, such as coastal archipelagos, is challenging. The wave spectrum, $E(f)$, contains a majority of the information about the wave field, and its properties have been studied for decades. Nevertheless, any systematic research into the wave spectrum in archipelagos has not been made. In this paper we present wave buoy measurements from 14 locations in

the Finnish archipelago. The shape of the wave spectrum showed a systematic transition from a single peaked spectrum, to a spectrum with a wide frequency range having almost constant energy. The exact shape also depended on the wind direction, since the fetch, island, and bottom conditions are not isotropic. The deviation from the traditional spectral form is strong enough to have a measurable effect on the definitions of the significant wave height. The relation between the two definitions in the middle of the archipelago was $H_{1/3} = 0.881 H_s$, but the ratio varied with the spectral width ($H_s$ was defined using the

variance). At this same location the average value of the single highest wave, $H_{max}/H_s$, was only 1.58. A wider archipelago spectrum was also associated with lower confidence limits for the significant wave height compared to the open sea (6 % vs. 9 %). The challenges regarding the instability of the peak frequency and the difficulties in finding a good "characteristic" frequency for an archipelago spectrum is discussed. The mean frequency, weighted with $E(f)^4$, is proposed as a compromise between stability, and bias with respect to the peak frequency.

## 1 Introduction

Since the 1950's the wave spectrum has been the central way to define the properties of random sea-surface wind waves (Pierson and Marks, 1952). Although the exact power law describing the high-frequency part of the spectrum is still an open question (e.g. Phillips, 1958; Toba, 1973; Kitaigorodskii, 1983; Kahma, 1981; Banner, 1990; Lenain and Melville, 2017), the central determining feature has been the location of the spectral maximum, which then consequently determines the total wave

energy (Hasselmann et al., 1973; Donelan et al., 1985). From a practical perspective of derived wave parameters the spectral features translate into the peak frequency, $f_p$, and the significant wave height, $H_s$. The evolution of these two parameters, as a function of the fetch and the wind speed, has been extensively studied by laboratory and field experiments (e.g. Pierson and Moskowitz, 1964; Toba, 1972; Hasselmann et al., 1973; Kahma, 1981; Donelan et al., 1985; Kahma and Calkoen, 1992).

In the coastal region waves are important for coastal constructions, erosion, small vessel safety, and biological processes.

Coastal waves deviate from deep water open-sea waves, but their exact properties depend on the shoreline structure. On sloping beaches the limitation by the water depth is a major factor shaping the wave properties through bottom friction, depth-induced





wave breaking, and shallow water non-linear wave interactions (SPM, 1984; Eldeberky, 1996). Coastal coral reefs shape the wave spectrum away from the deep-water form (Hardy and Young, 1996); the same thing can be said for tidal inlets where the waves are also affected by strong currents (van der Westhuysen et al., 2012).

One of the most complex nearshore conditions are found in coastal archipelagos where islands, the irregular shoreline, the
slanting fetch, and the decreasing depth affect the attenuation and local growth of the waves. Collections of large islands, in the scale of kilometres, can be found in e.g. the Gulf of Mexico outside of Louisiana, or between Vancouver and Seattle (the San Juan Islands). In Europe an example is the Aegean Sea, which separates parts of Turkey and Greece from the Mediterranean Sea. Denser archipelagos, where the island sizes are in the order of a hundreds of metres, are even more complex. An archipelago made up of a large magnitude of small islands have a strong effect on the wave field by attenuating the waves
and diffracting the remaining wave energy behind them. At the same time groups of islands practically define new fetches for local wave systems to grow from, thus giving birth to quite unique wave conditions. Examples of such archipelagos are the Thousand Islands at the US-Canadian border, or the coastline of Maine. In Europe dense coastal archipelagos can be found especially in the Baltic Sea, with examples being the Stockholm archipelago and the Archipelago Sea. Also the coastline near the Finnish capital, Helsinki, has a characteristic archipelago with heavy commercial and recreational marine traffic.

Although coastal archipelagos are usual—almost typical—in the Baltic Sea, there is a limited amount of data available on their effect on waves. Kahma (1979) presented measurements of wave spectra in the archipelago that showed an almost complete absence of the traditional spectral peak. Single measurements like these have proven the shape of the wave spectrum to differ significantly from both open sea observations and theoretical spectral models. Nevertheless, there exists no broader methodological study into the different spectral shapes and the transition between the two extremes, not least because of
the limited amount of available observational data. Efforts to simulate the wave field in the archipelagos have been made (Soukissian et al., 2004; Mazarakis et al., 2012; Tuomi et al., 2014; Anderson et al., 2015; Björkqvist et al., 2019), but while fairly successful, they are still no substitute for measurements.

This paper aims to fill the knowledge gap regarding the properties of the wave field inside dense coastal archipelagos. The study relied on spatially extensive wave buoy measurements; all data were collected in the Helsinki archipelago, which is
located in the Gulf of Finland, the Baltic Sea. The data and methods are introduced in Sect. 2, while Sect. 3 presents and quantifies the transformation of the mean spectral shape in the archipelago. Sect. 4 is dedicated to studying what implications the results have for the determination of derived wave parameters, such as the significant wave height, the maximum height of a single wave, and the peak frequency. One candidate for a "characteristic" frequency, suitable for a wide range of wave conditions, is proposed. We end the paper by discussing and concluding our findings.

## 2  Materials and methods

### 2.1  Wave measurements

We conducted wave measurements at 14 locations off Helsinki in the Gulf of Finland (GoF, Fig. 1). All observations were made with Datawell Directional Waveriders. Some of the data originate from smaller 40 cm GPS-based DWR-G4 buoys (henceforth



G4), which use the Doppler shift of the GPS-signal to measure the surface elevation. Measurements from larger (70 cm–90 cm) accelerometre based Mk-III and DWR4/ACM buoys (henceforth Mk-III and DWR4) were available from two operational wave buoys: one is located in the centre of the GoF, while the other is deployed in the middle of the archipelago outside of the island Suomenlinna (site T2, Fig. 1).

The nearshore measurements with the G4 buoys—conducted as part of a commissioned work by the City of Helsinki—were made for about a month at each location between 2012 and 2014 (Table 1). The shortest deployment time was 11 days (at Länsikari) and the longest 39 days (at Isosaari). The measurements were made between August and November to capture the harshest wave conditions before the areas froze. While 12 out of the 14 locations were made with the smaller G4 buoys, most of the data originate from the long time series of the operational Mk-III and DWR4 wave buoys at GoF and Suomenlinna. Data

from the Suomenlinna wave buoy were available from 2016–2018. Operational wave measurements from the GoF have been conducted for every year in this study (2012–2018), but the results are based only on the years 2016–2018 to coincide with the Suomenlinna data. Table 1 lists the measurement sites and their basic wave statistics.

Both the G4 and the Mk-III use a sampling frequency of 1.28 Hz and calculates the spectrum up to 0.58 Hz. A DWR4 buoy was used at the GoF site in 2018 and part of the year 2016. The DWR4 has a sampling frequency of 2.56 Hz and the 90 cm

version calculates the spectrum up to 1 Hz. Nevertheless, only data up to 0.58 Hz were used to keep all results comparable, since the change in upper frequency would affect the calculations of higher spectral moments.

For the G4 buoys the low-frequency artefacts, which have later been identified to be caused by the filter response to a missing GPS-signal, were corrected following Björkqvist et al. (2016). Since the authors found that the correction can affect the high frequency part of the spectrum, the corrected spectrum was only used for frequencies below $0.8f_m$, where $f_m$ is the mean

frequency (see Eq. (4)). For frequencies above $1.2f_m$ Hz the original spectrum was used, while a linear combination was used for intermediate frequencies to avoid discontinuities.

## 2.2 Wind measurements

We used wind measurements from two of the Finnish Meteorological Institutes operational automatic weather stations. Harmaja (measuring height 17.5 m) is located less than 10 km from the Helsinki shoreline, about 5 km from the Suomenlinna wave

buoy (Fig. 1). The Kalbådagrund station (measuring height 31.8 m) is located in the middle of the Gulf of Finland, about 20 km east of the operational GoF wave buoy. Both stations have been active during the entire period of the study, but there are long gaps in the Kalbådagrund data in August and September 2018.

The weather stations provided the wind speed and direction averaged over 10 minutes. We calculated the 30 minute averages from these data to coincide with the length of the wave buoy time series.



**Figure 1.** The bathymetry and the measurement locations. GoF (green) is the open sea wave buoy. The stations are devided into groups: Outer archipelago (O1–O3), Transition zone (T1–T3), Inner archipelago (I1–I3), and Sheltered archipelago (S1–S4). The plus (+) marks the Harmaja weather station. The Kalbådagrund weather station is outside of the map.





## 2.3 Wave parameters and definitions

### 2.3.1 Spectral wave parameters

The wave buoys calculated the wave spectrum $E(f)$ ($\mathrm{m^2\ Hz^{-1}}$), where $f$ (Hz) is the frequency. The $n$-th order moment of the wave spectrum is

$$m_n = \int f^n E(f)\,\mathrm{d}f. \tag{1}$$

Using these moments we defined most of the relevant wave parameters. The significant wave height $H_{m_0}$ was defined as

$$H_{m_0} = 4\sqrt{m_0}. \tag{2}$$

A spectral version of the zero down-crossing period was defined as:

$$T_{m_{02}} = \sqrt{\frac{m_0}{m_2}}, \tag{3}$$

10     while the mean frequency was given by

$$f_m = \frac{m_1}{m_0}. \tag{4}$$

We defined the peak frequency as

$$f_p = \arg\max_f E(f), \tag{5}$$

i.e. the frequency where the wave spectrum has its maximum value. Because of the discrete frequency intervals and statistical

15     variations in the spectrum, several methods for calculation the peak frequency have been proposed. In this paper we calculated $f_p$ using a parabolic fit. We, however, also applied an integrated definition (Young, 1995):

$$f_p^q = \frac{\int f E(f)^q\,\mathrm{d}f}{\int E(f)^q\,\mathrm{d}f}, \tag{6}$$

where $q$ in a positive integer. Note, that for $q = 1$ Eq. (6) equals the mean period $f_m$ given by Eq. (4).



### 2.3.2 Wave parameters from time series

We also determined wave parameters directly from the 30 minute vertical displacement time series, $\eta(t)$.

The significant wave height, $H_s$, was defined

$$H_s = 4\,\sigma(\eta), \tag{7}$$

where $\sigma$ is the standard deviation. This definition is identical to Eq. (2) with the exception of the statistical variability introduced by the window tapering of the time series before calculating the spectrum.

The traditional definition of the significant wave height is the mean height of the highest one third of the individual waves in the time series. To distinguish it from the significant wave height calculated using the variance, we will denote this parameter $H_{1/3}$. The individual waves were determined between two zero down-crossings, sorted in descending orders, and the mean was calculated as

$$H_{1/3} = \frac{1}{N/3} \sum_{i=1}^{N/3} H_i, \tag{8}$$

where $H_i$ is the height of a single wave and $N$ is their total number.

The zero down-crossing period, $T_z$, was calculated as

$$T_z = \frac{T}{N} \tag{9}$$

where $T$ is the length of the time series $\eta(t)$.

Assuming that $\eta(t)$ is Gaussian, $T_z = T_{m_{02}}$, and assuming that $H_i$ are Rayleigh distributed, $H_s = H_{m_0} = H_{1/3}$.

### 2.3.3 Spectral width parameters

Several parameters to quantify the spectral width have been proposed. The width parameter $\varepsilon$ (Cartwright and Longuet-Higgins, 1956) depends on the fourth moment, $m_4$, and is therefore sensitive to noise in the higher frequencies.

Longuet-Higgins (1975) defined a spectral width parameter, $\nu$, as

$$\nu = \sqrt{\frac{m_0 m_2}{m_1^2} - 1}, \tag{10}$$

which, to a certain degree, suffers from similar issues as $\varepsilon$. Two other width (narrowness) parameters were used in this study. The first was the $\kappa^2$ parameter of Battjes and van Vledder (1984):

$$\kappa^2 = \frac{1}{m_0{}^2} \left( \left[ \int\limits_0^\infty E(f) \cos(\frac{2\pi f}{f_{m_{20}}})\,\mathrm{d}f \right]^2 + \left[ \int\limits_0^\infty E(f) \sin(\frac{2\pi f}{f_{m_{20}}})\,\mathrm{d}f \right]^2 \right), \tag{11}$$




where $f_{m_{20}} = T_{m_{02}}^{-1} = \sqrt{m_2/m_0}$. This parameter was used as the main way to quantify the width of different spectral shapes. The other parameter was the Goda peakedness parameter (Goda, 1970), defined as:

$$Q_p = \frac{2}{m_0^2} \int\limits_0^\infty f E(f)^2 \mathrm{d}f. \tag{12}$$

The Goda peakedness parameter was needed in the definition of the Benjamin-Feir Index (BFI, Janssen, 2003), which is

essentially the wave steepness divided by the spectral width. We used the BFI to quantify its possible connection with single waves that are high compared to the significant wave height, i.e. so called "rogue waves", where $H/H_s > 2$. The formulation given by Serio et al. (2005) is:

$$\mathrm{BFI} = \sqrt{m_0} k_p Q_p \sqrt{2\pi} \alpha_0 \sqrt{\frac{|\beta|}{\alpha}}, \tag{13}$$

where the peak wavenumber, $k_p$, is estimated from $f_p$ using linear wave theory. The coefficients $\alpha_0, \alpha$, and $\beta$ depend on the

dimensionless depth, $k_p h$. Their exact expressions are given by e.g. Serio et al. (2005).

## 2.4    Choosing wave spectra and spectral scaling

Data were available from more locations than the 14 presented in this paper (Kahma et al., 2016). We, however, excluded some stations based on: i) very small maximum wave heights, meaning that the wave buoy was often unable to measure the entire spectrum, ii) the location was not even remotely exposed to open-sea waves (a determining factor for the archipelago type

spectrum), or iii) the location was too exposed to external disturbances, such as wave reflection or ship wakes.

As a loose definition of well defined wave conditions we used the $80^{th}$ percentile of the significant wave height as a cut-off for each location. In addition we used a cut-off of $U \geq 5 \ \mathrm{ms}^{-1}$, where $U$ is the 30 minute average wind speed. For the GoF wave buoy we used the Kalbådagrund data, while Harmaja wind data were used for all other locations. For the nearshore locations only onshore winds were considered ($70° \leq U_d \leq 250°$), while no restrictions on the wind direction was set for the

GoF. Henceforth, we will call this data set the P80 data set. For the GoF data from the years 2016–2018 only were included to keep the measurement period comparable with the Suomenlinna observations.

The choice of the $80^{th}$ percentile was a compromise between: i) removing the smallest wave heights, e.g. $H_s < 0.5$ at Suomenlinna, since they are bound to be noisy, and ii) not excluding too much data from the limited data sets available from the short measurements. Using a different cut-off for the significant wave height ($60–90^{th}$ percentile) resulted in very similar

results. We also tried setting restrictions with respect to the steadiness of the wind direction and the wind speed, but imposing these additional restrictions resulted in very similar results and identical conclusions. To avoid adding unnecessary complexity, these additional constraints were dropped. Also, some of the highest wave heights at the GoF buoy were measured during a time where no wind data were available (August–September 2018). Cases with missing wind data were therefore included if they fulfilled the conditions set for the significant wave height.





Because the short waves are generated by the shortest fetch, they are least affected by the varying spectral shape inside the archipelago. The chosen spectra were therefore normalised using the values at the high frequencies ($f > 0.4$ Hz). The scaled spectra were calculated as:

$$\tilde{E}(f) = \frac{E(f)}{\beta}, \tag{14}$$

where

$$\beta = \langle E(f) \rangle_{f > 0.4 Hz} \tag{15}$$

and the brackets signifies a mean value over the frequencies $f > 0.4$ Hz. The exact value of the lower frequency, $f_0 = 0.4$ Hz is unimportant as long as the spectrum follows some kind of power law for $f > f_0$. Since we had no reliable way of determining the starting point of the typical $f^{-4}$ power law of the spectral tail, we chose a frequency that is sufficiently high for the strong wind condition that are represented in the P80 data set.

The frequencies were then normalised with respect to the mean frequency and the spectra, $\tilde{E}$, were interpolated to a common set of dimensionless frequencies $\tilde{f} = f/f_m$. This resulted in the final scaled spectra $\tilde{E}(\tilde{f})$. The mean frequency was chosen instead of the peak frequency because it is more stable. Using this same data set Björkqvist et al. (2019) found that the peak frequency can be highly noisy in the archipelago, and is therefore not usable to scale the spectra. The choice of good "characteristic" frequency for archipelago conditions will be studied in Sect. 4.5.

## 2.5 Determining groups

The 13 measurement stations in the archipelago were divided into four groups based on an visual estimation of the geographical conditions. The attenuation coefficients for the significant wave height compared to the GoF wave buoy were used as a crude check to ensure that the visual determination of the amount of sheltering was reasonable. The attenuation coefficients, $K$, were determined by a linear fit using the effective variance method (Orear, 1982). The different groups, visible in Fig. 1, can be described as follows:

**Outer archipelago (O1–O3)**: The locations are not inside the archipelago, but the effect of the finite depth and/or the limited fetch caused by the irregular shoreline might be visible in the wave spectrum. Although the O2 station (Harmaja) seems to be very exposed, Björkqvist et al. (2017) have shown that the wave field here is already restricted by the peninsula of Porkkala for south-westerly winds. The attenuation coefficients for the significant wave height were $K = 0.6$–$0.7$.

**Transition zone (T1–T3)**: The sheltering of the islands play a significant role in shaping the waves field, but the longer waves propagating from the GoF are still somewhat dominant. The attenuation coefficients for the significant wave height were $K = 0.4$.

**Inner archipelago (I1–I3)**: There is a clearly defined local fetch, but there is still a significant contribution from longer propagating waves. The attenuation coefficients for the significant wave height were $K = 0.2$.

**Sheltered archipelago (S1–S4)**: These locations should be dominated by the locally generated waves. Residuals of longer waves can, however, still be present. The attenuation coefficients for the significant wave height were very small ($K < 0.10$).





**Table 1.** The measurement time and depth at the different sites (see Fig. 1 for an overview). The mean and $80^{th}/20^{th}$ percentiles are shown for the significant wave height ($H_{m_0}$), the peak frequency ($f_p$), and the mean frequency ($f_m$). The most probable value of the mean direction at the spectral peak ($\theta_p$) is also shown. All available spectra were used to compile these statistics.

| | | | mean | $P_{80}$ | mean | $P_{20}$ | mean | $P_{20}$ | most probable |
|---|---|---|---|---|---|---|---|---|---|
| Name (Code) | days | depth | $H_{m_0}$ (m) | | $f_p$ (Hz) | | $f_m$ (Hz) | | $\theta_p$ (deg) |
| Gulf of Finland (GoF) | 790 | 62 m | 0.80 | 1.22 | 0.23 | 0.16 | 0.29 | 0.23 | 245–255 |
| **Outer archipelago** | | | | | | | | | |
| Harmaja (O1) | 30 | 29 m | 0.61 | 0.92 | 0.24 | 0.18 | 0.29 | 0.25 | 215–225 |
| Isosaari (O2) | 38 | 7 m | 0.34 | 0.51 | 0.27 | 0.19 | 0.31 | 0.26 | 205–215 |
| Berggrund (O3) | 11 | 27 m | 0.45 | 0.59 | 0.25 | 0.18 | 0.30 | 0.26 | 215–225 |
| **Transition Zone** | | | | | | | | | |
| Lansikari (T1) | 10 | 10 m | 0.49 | 0.65 | 0.24 | 0.20 | 0.30 | 0.27 | 185–195 |
| Suomenlinna (T2) | 662 | 22 m | 0.33 | 0.50 | 0.29 | 0.19 | 0.34 | 0.29 | 195–205 |
| Itä-Villinki (T3) | 31 | 9 m | 0.28 | 0.49 | 0.24 | 0.18 | 0.31 | 0.27 | 135–145 |
| **Inner archipelago** | | | | | | | | | |
| Hernesaari (I1) | 31 | 13 m | 0.20 | 0.29 | 0.28 | 0.22 | 0.33 | 0.30 | 155–165 |
| Ruumiskari (I2) | 28 | 12 m | 0.24 | 0.40 | 0.30 | 0.19 | 0.35 | 0.31 | 145–155 |
| Jätkäsaari (I3) | 30 | 13 m | 0.17 | 0.27 | 0.30 | 0.22 | 0.35 | 0.31 | 165–175 |
| **Sheltered archipelago** | | | | | | | | | |
| Koivusaari (S1) | 27 | 5 m | 0.05 | 0.08 | 0.42 | 0.32 | 0.41 | 0.38 | 175–185 |
| Ramsinniemi (S2) | 31 | 9 m | 0.05 | 0.08 | 0.45 | 0.33 | 0.42 | 0.38 | 115–125 |
| Vuosaaren satama (S3) | 14 | 8 m | 0.02 | 0.04 | 0.48 | 0.39 | 0.43 | 0.41 | 155–165 |
| Talosaari (S4) | 33 | 7 m | 0.02 | 0.03 | 0.49 | 0.44 | 0.45 | 0.41 | 195–205 |

The common denominator throughout the archipelago is that the waves travel slower than the wind. Thus, the longer waves propagating from the open sea are not swell. In this paper we will show that the sheltering effect of the archipelago is a continuum and several reasonable classifications could therefore be made. The main purpose of the classification was to make the results more presentable.





## 3 The archipelago spectrum

### 3.1 Transition from peaked to flat spectra

The main result of this section is that the wave spectrum transitioned from a, more traditional, single peaked spectrum to a flat spectrum inside the archipelago. The transition was continuous, as readily seen in Fig. 2. In the Outer archipelago (black) the mean spectrum had a very similar shape to the open sea conditions observed at the GoF wave buoy (green). Namely, it had a single peak even if it lacked the overshooting of an even more peaked fetch-limited spectrum.

When moving closer towards the coast the spectral shape started to flatten out in the Transition zone (blue). Länsikari (T1) and Suomenlinna (T2) are located very close to each other (Fig. 1), and the similar spectral shapes gives confidence that we captured the shape of the mean spectrum even with the shorter measurement time series. Although the mean spectral profiles in the Transition zone were still peaked, the rear face of the spectrum was starting to collapse. In contrast to the Outer archipelago, the mean spectra in the Transition zone decayed slower than $\tilde{f}^{-4}$ just above the spectral peak. Especially T3 was showing a clear collapse towards a flat spectral shape.

In the Inner archipelago (red) the spectral shape had collapsed around the peak and exhibited a constant energy in a broad frequency interval ($0.6 f_m \leq f \leq 1.1 f_m$). Even if the peak frequency could be reliably determined—which is challenging because of the statistical variability—it is obvious that it wouldn't characterise the spectrum in a similar fashion as the spectral peak in e.g. the Outer archipelago. There were, however, small low-frequency peaks, most notably at Jätkäsaari (I3). These peaks were caused by refracted, narrow banded, waves and are therefore expected to be specific to the bathymetrical conditions of the area.

In the Sheltered archipelago (magenta) even these attenuated low-frequency peaks were no longer visible. The mean spectrum at Koivusaari (S1) was still flat (in a similar fashion to sited I1–I3), but for the other sheltered locations the spectrum were almost transitioned back to a single peaked shape—the local fetch was starting to dominate over the very attenuated longer waves. The tail of the spectrum was not determined reliably, since these short waves were often not captured by the wave buoy.

### 3.2 Quantifying the spectral width

We quantified the change in width (or more exactly, narrowness) of the spectrum using the $\kappa^2$ narrowness parameter (Eq. (11)) of Battjes and van Vledder (1984). The mean width of the wave spectrum changed when moving into the archipelago, with $\kappa^2$ being 0.03–0.07 in the Inner archipelago (signaling a wide spectrum), while being 0.18–0.19 in the Outer archipelago (Table 2). The value in the Outer archipelago was close to the one at the open sea location in the GoF. As an example, we also calculated the $\kappa^2$ parameter for the single storm spectrum of the measured maximum 5.2 m significant wave height during the easterly Antti storm in 2012. The value of $\kappa^2 = 0.33$ was higher than the average value at the GoF, but this storm spectrum is affected by the narrow fetch geometry of the GoF, which leads to a less peaked spectrum (Pettersson, 2004). Higher values (up to $\kappa^2 = 0.46$) were found at the GoF.

The spectral width in the Transition zone is in between those of the Outer and Inner archipelago ($0.08 \leq \kappa^2 \leq 0.15$). The almost identical width of T1 and T2, and the wider shape of T3, are in good agreement with what was determined visually





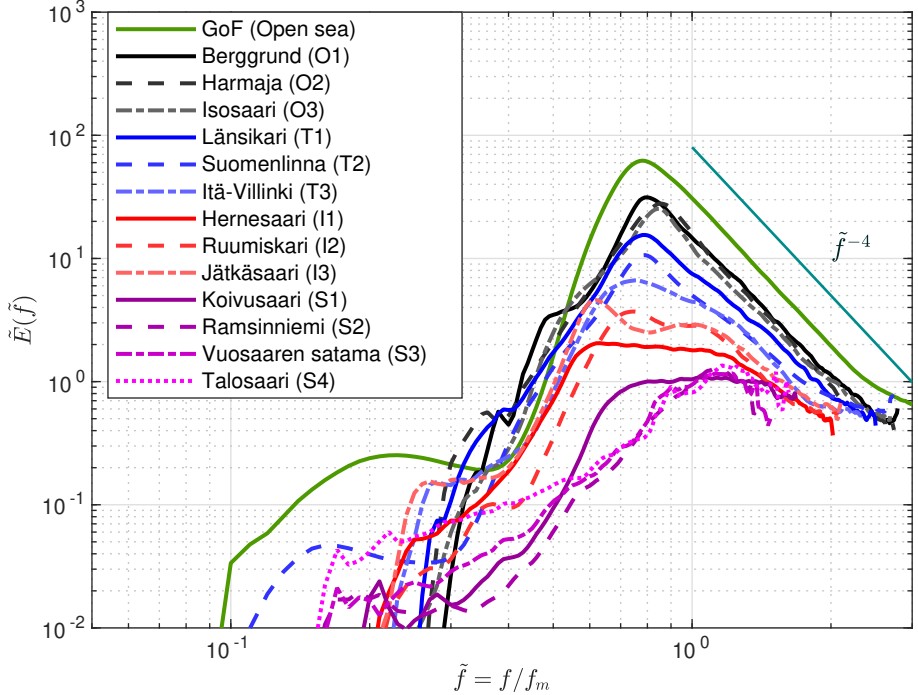

**Figure 2.** The mean wave spectra divided into the open sea and four archipelago areas: Outer archipelago (O1–O3), Transition zone (T1–T3), Inner archipelago (I1–I3), and Sheltered archipelago (S1–S4). An overview of the locations is found in Fig. 1.

from Fig. 2. The $\kappa^2$ values for the Sheltered archipelago sites (S1–S4) were variable, which was a consequence of the wave buoys issues with resolving the entire spectrum.

Quantifying the spectral width is not a trivial matter, but the good agreement between the $\kappa^2$ parameter and the obvious visual changes suggests that the parameter is applicable over a wide range of conditions.

5  ### 3.3  Directional dependence

Although the mean spectral profiles were shown to change when moving through the archipelago towards the shore, the spectral shape also varied with the wind direction because of the anisotropic fetch conditions. We used the wind direction because the instability of the spectral peak at Suomenlinna made it hard to define the dominant wave direction. The wave direction at the GoF buoy, again, collapses to be aligned with the gulf, thus causing a misalignment of up to 50° between the direction

10 of the wind and the dominant waves (Pettersson et al., 2010). Nevertheless, the local fetch at Suomenlinna would still vary significantly within this large wind sector. Suomenlinna is the only location in the archipelago with enough data to partition it further with respect to the wind direction. This section will therefore be based on data from Suomenlinna only.

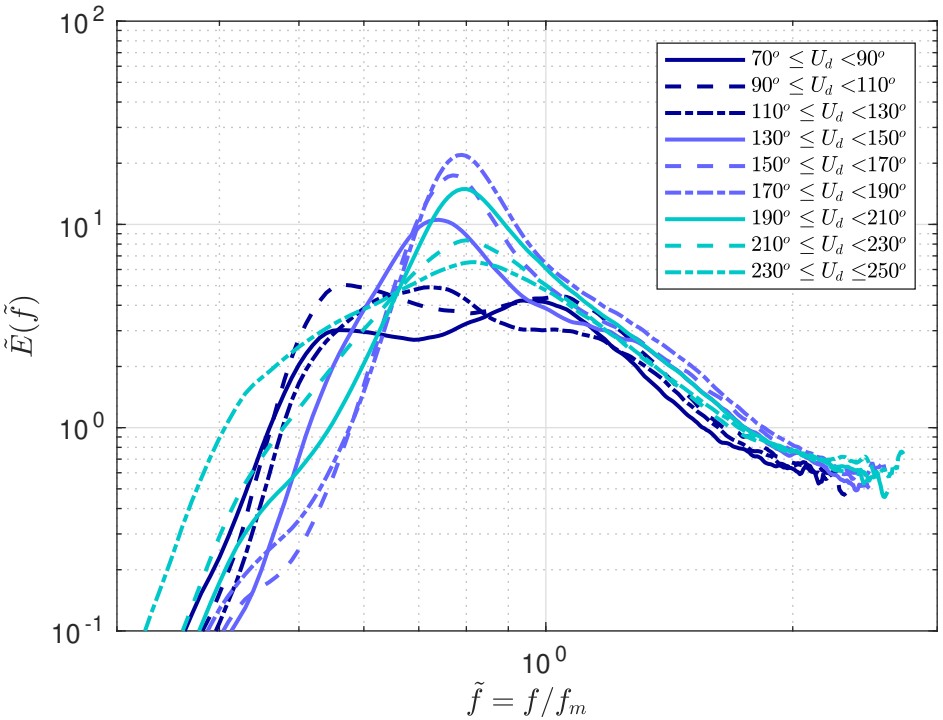

**Figure 3.** The mean wave spectra at Suomenlinna (T2) as a function of the wind direction.

The most peaked spectra at Suomenlinna are generated by southerly winds (Fig. 3), since only small islands are blocking the wave propagation in this direction (Fig. 1). For easterly winds the spectral shape is flat, resembling the profile characteristic for the Inner archipelago (I1–I3, Fig. 2). Such a variation was identified already by Björkqvist et al. (2019) when studying wave model performance against the Suomenlinna wave data, but our results show a systematic behaviour. Importantly, the eastern wind directions showed a very flat mean spectrum even though the average shape over all wind directions was peaked. This discrepancy is explained by easterly winds not being dominant ($45° \leq U_d \leq 135°$ 10 % of the times). Nevertheless, strong easterly winds are possible in the GoF; the maximum significant wave height of 5.2 m at the GoF wave buoy has been measured twice, both during south-westerly winds (in 2001, Tuomi et al., 2011) and easterly winds (in 2012, Pettersson et al., 2013).

The spectral shapes at Suomenlinna covers the more peaked shape of the Outer archipelago sites and the flat shape of the Inner archipelago sites (Fig. 2). In some sections we will therefore use the long, three year, time series at Suomenlinna to study how the different spectral shapes affect certain properties of the wave field. The Suomenlinna data was used instead of data from different sites to avoid variations in water depth and issues with resolving the spectral tail.





## 4 Implications

### 4.1 Confidence limits of significant wave height

The confidence limits of observed wave spectra follow a $\chi_k^2$-distribution, where $k$ is the degrees of freedom determined by the number of averaged elementary bins. The confidence limits of the spectrum propagate to its integral, which is also the total

variance of the wave field, $m_0$. By Eq. (2) the confidence limits of the observed significant wave height follow from those of $m_0$.

The final degrees of freedom (d.o.f.) of the integral of a measured spectrum depend on the shape of the spectrum in the following way (Donelan and Pierson, 1983):

$$\text{d.o.f.}(m_0) = \frac{k\left[\sum_{i=1}^{N} E(f)\right]^2}{\sum_{i=1}^{N}\left[E(f)\right]^2}. \tag{16}$$

It immediately follows that d.o.f.$(m_0) = kN$ for a white noise spectrum ($E(f) \equiv$ const.), while d.o.f.$(m_0) = k$ for an infinitely peaked spectrum ($E(f) = \delta(f - f_p)$). Thus, a broader spectrum will have a more d.o.f., leading to smaller confidence limits for the significant wave height.

In Sect. 3.2 we quantified the spectral width using the $\kappa^2$ parameter. The change in spectral width should also be seen in the d.o.f. calculated at the different locations. This was, indeed, the case: the d.o.f. in the Inner archipelago were roughly 500–600,

while the corresponding values in the Outer archipelago were around 300 (Table 2). The Transition zone, again, had values falling in between the Inner and Outer archipelago, with the d.o.f at site T3 being closest to the Inner archipelago. In the open sea location (GoF) the d.o.f. of the variance were lower than anywhere in the archipelago, and the low d.o.f. of the single GoF storm spectrum went hand-in-hand with the large $\kappa^2$ value.

The increase in the d.o.f. in the archipelago had a direct implication for the confidence limits of the significant wave height:

the confidence limits at the GoF wave buoy were 50 % larger than at the Inner archipelago points (Table 2). The confidence limits of the single storm spectrum was twice that of the average value in the Inner archipelago (12 % vs 6 %).

Because the spectral shape depended strongly on the wind direction at Suomenlinna (T2), the confidence limits for easterly winds were close to those of the Inner archipelago, while south-westerly—and especially southerly—winds resulted in confidence limits in line with the open sea (Fig. 4 a). By comparing the d.o.f. to the $\kappa^2$ parameter it is obvious that they were both

quantifying a similar property of the spectrum (Fig. 4 b). The correlation between these two parameters were $r = -0.94$, and also the Goda peakedness parameter (Eq. (12)) was correlated ($r = -0.86$) with the d.o.f. of the spectrum (Fig. 4 c).

The correlation between the d.o.f. and the spectral width parameter $\nu$ by Longuet-Higgins (1975) was only $r = 0.2$ (Fig. 4 d), and the correlation was equally low for the spectral width parameter $\varepsilon$ proposed by Cartwright and Longuet-Higgins (1956) (not shown).


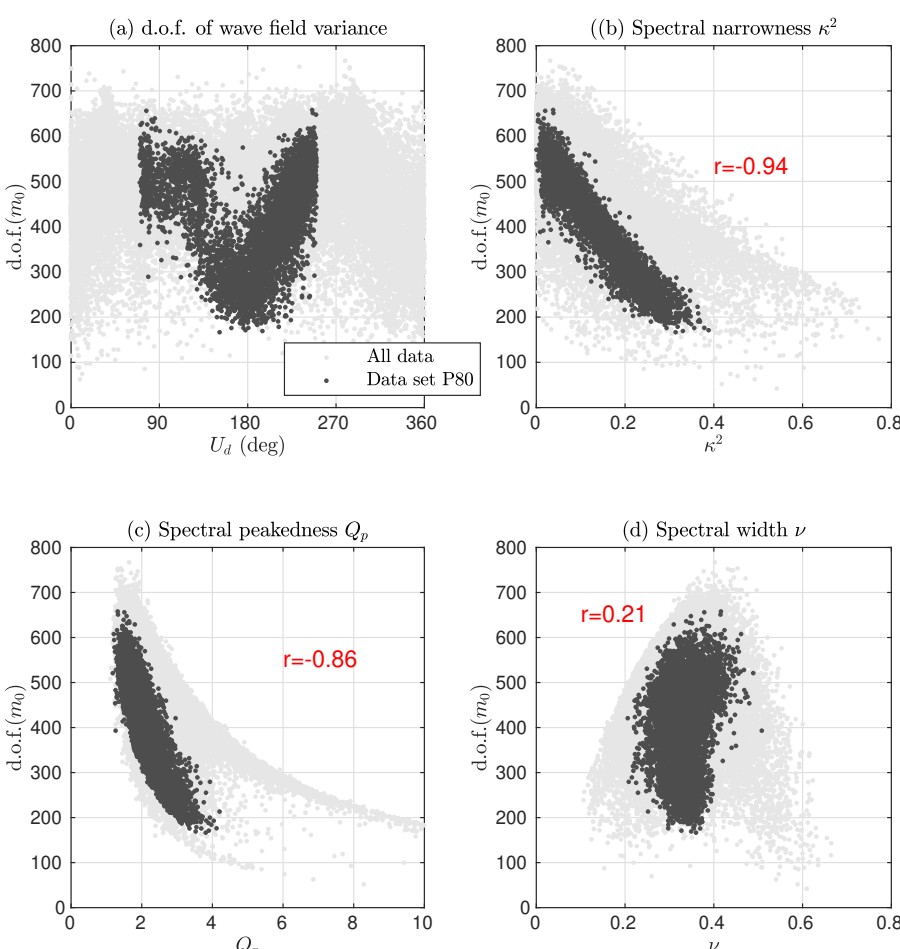

**Figure 4.** The degrees of freedom of the Suomenlinna spectra as a function of the wind direction (a), and compared to different spectral width/narrowness parameters (b–d).

## 4.2 The significant wave height: $H_{1/3}$ vs. $H_s$

The significant wave height is the most central and widely used wave parameter. Still, it can be defined in a couple of different ways. The connection between the definition using the mean height of the highest one third of the single waves, and the definition based on the variance of the vertical displacement was made based on the assumption of a narrowbanded spectrum,





**Table 2.** Mean values of the spectral narrowness parameter ($\kappa^2$), and the number of degrees of freedom for the variance of the wave field. The mean values are calculated only from the data set P80. In the confidence limits $\hat{H}_{m_0}$ denotes a sample from a wave field with a significant wave height of $H_{m_0}$.

| Name (Code) | $<\kappa^2>$ | $<\text{d.o.f.}(m_0)>$ | 95 % confidence limits |
|---|---|---|---|
| Single storm spectrum (GoF) | 0.33 | 132 | $0.88 < \hat{H}_{m_0}/H_{m_0} < 1.12$ |
| Gulf of Finland (GoF) | 0.22 | 234 | $0.91 < \hat{H}_{m_0}/H_{m_0} < 1.09$ |
| **Outer archipelago** | | | |
| Harmaja (O1) | 0.19 | 316 | $0.92 < \hat{H}_{m_0}/H_{m_0} < 1.08$ |
| Isosaari (O2) | 0.19 | 323 | $0.92 < \hat{H}_{m_0}/H_{m_0} < 1.08$ |
| Berggrund (O3) | 0.18 | 309 | $0.92 < \hat{H}_{m_0}/H_{m_0} < 1.08$ |
| **Transition Zone** | | | |
| Lansikari (T1) | 0.15 | 370 | $0.93 < \hat{H}_{m_0}/H_{m_0} < 1.07$ |
| Suomenlinna (T2) | 0.14 | 410 | $0.93 < \hat{H}_{m_0}/H_{m_0} < 1.07$ |
| Itä-Villinki (T3) | 0.08 | 454 | $0.93 < \hat{H}_{m_0}/H_{m_0} < 1.06$ |
| **Inner archipelago** | | | |
| Hernesaari (I1) | 0.07 | 485 | $0.94 < \hat{H}_{m_0}/H_{m_0} < 1.06$ |
| Ruumiskari (I2) | 0.05 | 524 | $0.94 < \hat{H}_{m_0}/H_{m_0} < 1.06$ |
| Jätkäsaari (I3) | 0.03 | 577 | $0.94 < \hat{H}_{m_0}/H_{m_0} < 1.06$ |
| **Sheltered archipelago** | | | |
| Koivusaari (S1) | 0.13 | 496 | $0.94 < \hat{H}_{m_0}/H_{m_0} < 1.06$ |
| Ramsinniemi (S2) | 0.35 | 367 | $0.93 < \hat{H}_{m_0}/H_{m_0} < 1.07$ |
| Vuosaaren satama (S3) | 0.27 | 394 | $0.93 < \hat{H}_{m_0}/H_{m_0} < 1.07$ |
| Talosaari (S4) | 0.27 | 347 | $0.93 < \hat{H}_{m_0}/H_{m_0} < 1.07$ |

deep water, and that the height of single waves are Rayleigh distributed with the parameter $\sqrt{4m_0}$. These conditions lead to a proportionality constant of 4 in the equality

$$H_{1/3} = 4\sqrt{m_0} = 4\sigma(\eta). \tag{17}$$

Studies have, however, shown that the assumption of a Rayleigh distribution (with a parameter $\sqrt{4m_0}$) for the height of in-
5 dividual waves predicts higher values of $H_{1/3}/H_s$ compared to observations (Forristall, 1978; Longuet-Higgins, 1980; Casas-Prat and Holthuijsen, 2010). The discrepancy have been solved e.g. by assuming a Weibull distribution (Forristall, 1978), or by





simply scaling the Rayleigh parameter as $\alpha\sqrt{4m_0}$ (Longuet-Higgins, 1980). The use of a scaled Rayleigh distribution modifies Eq. (17) to

$$H_{1/3} = 4\alpha\sqrt{m_0}. \tag{18}$$

Longuet-Higgins (1980) determined $\alpha$ as a function of the spectral width:

$$5 \quad \alpha = \sqrt{1 - \left(\frac{\pi^2}{8} - \frac{1}{2}\right)\nu^2}, \tag{19}$$

where $\nu$ is the spectral width parameter of Longuet-Higgins (1975) (Eq. (10)). Since the original derivation of Eq. (17) assumed a narrowbanded spectrum with symmetrical Gaussian water level displacements, we expected that the two definitions for the significant wave height would vary even more in the archipelago than previously observed for open sea conditions.

We determined the fit between $H_{1/3}$ (Eq. (8)) and $H_s$ (Eq. (7)) that were calculated from the vertical displacement time series. The fit to the Suomenlinna P80 data set was $H_{1/3} = 0.881H_s$ (Fig. 5 a), which is a stronger deviation from Eq. (17) than found by previous studies (Table 3). The coefficient depended on the wind direction in a similar fashion as the spectral shape shown in Fig. 3; the more peaked spectral shapes of the southerly winds resulted in a proportionality constant of 0.90, while the corresponding value for the flat easterly spectra was around 0.86 (Fig. 5 c).

Vandever et al. (2008) found that $H_{1/3}/H_s$ depended on the spectral width and determined a best fit of $H_{m_0}/H_{1/3} = 0.996 + 0.181\nu$ from Doppler wave gauge data. We note that calculating the ratio $\alpha$ from Eq. (19) using $\nu$, as proposed by Longuet-Higgins (1980), increased the disagreement with our data for both Suomenlinna and GoF (Table 3). The value determined empirically by Longuet-Higgins (1980) using the data of Forristall (1978) (0.925) was, however, in very close agreement with 0.927 determined from our GoF data. The issue might have been caused by the reliable determination of $\nu$; the mean value of $\nu = 0.36$ (GoF) is lower than $\nu = 0.41$–$0.83$ reported by Vandever et al. (2008), although their data had swell present. We instead determined a linear fit using the narrowness parameter $\kappa^2$ and our Suomenlinna data, which resulted in $H_{1/3}/H_s = 0.85 + 0.15\kappa^2$ (Fig. 6 a). For an infinitely narrow spectrum ($\nu = 0$, $\kappa^2 = 1$) both fits result in approximately unity, which is in accordance with the narrowbanded assumption used to derive Eq. (17).

Even for the southerly waves at Suomenlinna $H_{1/3}/H_s$ was no higher than 0.90. It is therefore possible that the finite water depth (22 m at Suomenlinna) affected the results to a certain degree. The ratio $H_{1/3}/H_s$, however, showed a *negative* correlation ($r = -0.52$) with the dimensionless depth, $k_p h$, meaning that the largest deviations from deep water values are found for the cases where the water is deepest (relative to the waves). This is the opposite of what we would expect if the deviation from Eq. (17) was indeed caused by the finite depth effects. The apparent correlation was created by more sheltering simultaneously leading to both shorter waves (i.e. higher $k_p h$) and a wider wave spectrum. The wider spectrum can then explain the discrepancy through the $\kappa^2$ parameter as outlined above. We concluded that the deviation from Eq. (17) was mainly caused by the spectral shape, not the finite depth at the measurement site.

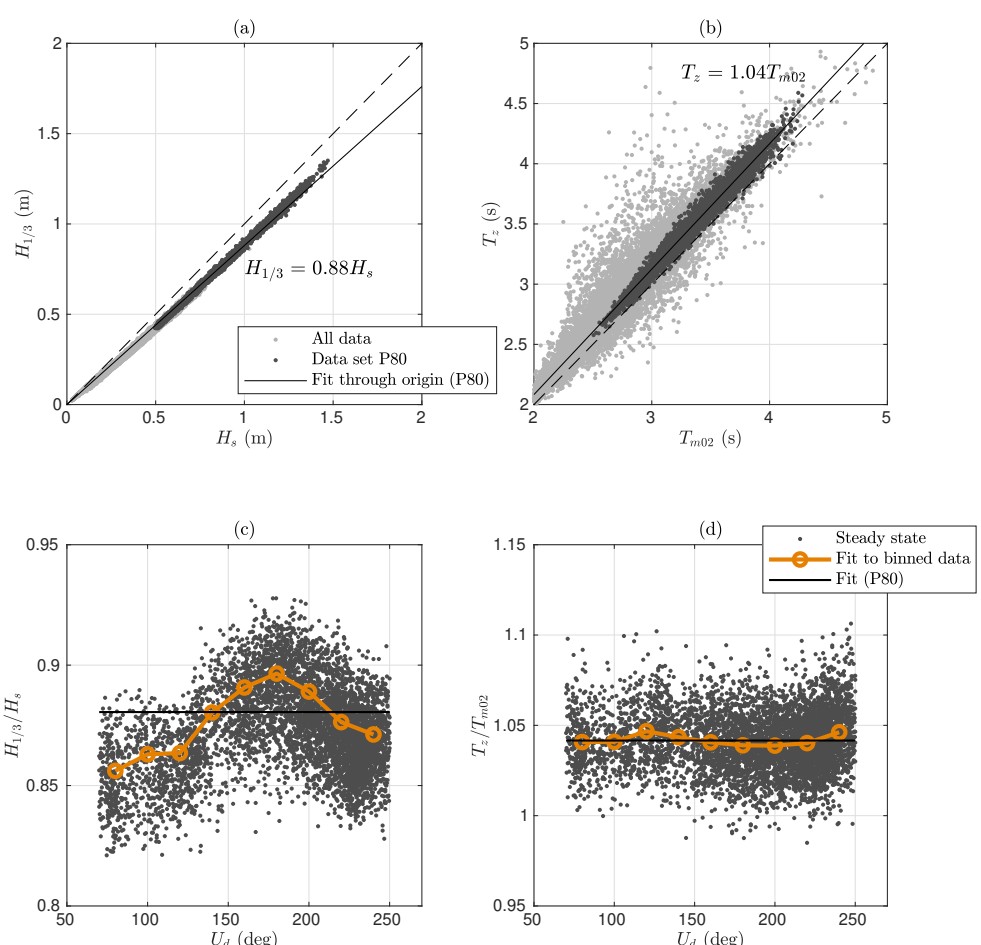

**Figure 5.** Comparison of $H_{1/3}$ and $T_z$ with respect to $H_s$ and $T_{m02}$ at Suomenlinna. In panels c) and d) the ratios are given as a function of the wind direction.

### 4.3 Single wave statistics: $H_{max}/H_s$

The highest expected single wave is of often of considerable interest, and usually this single wave is given relative to the significant wave height. The estimate is made based on the assumption that the height of the single waves are either Rayleigh or Weibull distributed. The estimated highest single wave thus depends on the assumed distribution and the number of waves encountered during the measurement period ($N$).

5     We determined the highest single wave from the vertical displacement time series of the P80 data sets. For the GoF the connection between the single wave height and the significant wave height was determined to be $H_{max} = 1.61 H_s$ using a





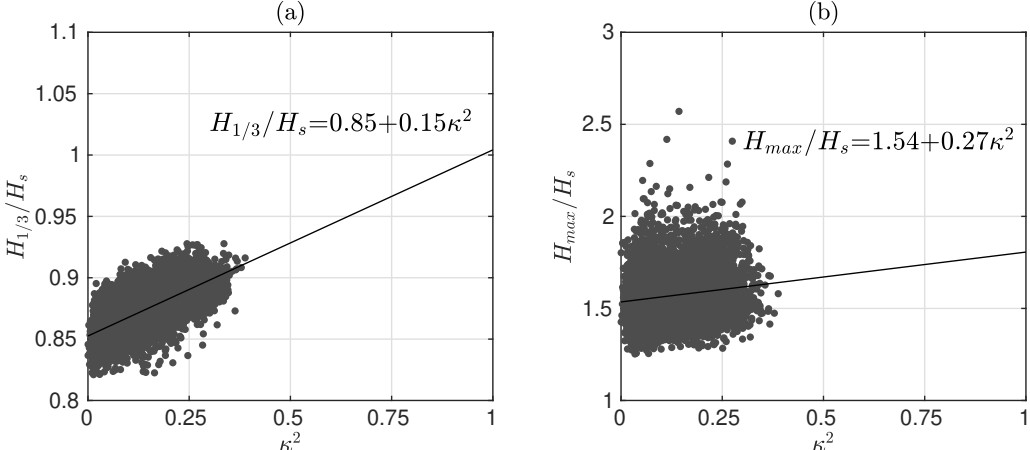

**Figure 6.** $H_{1/3}$ (a) and $H_{max}$ (b) at Suomenlinna relative to the significant wave height. The ratios are plotted against the spectral narrowness $\kappa^2$. For an infinitely narrow spectrum ($\kappa^2 = 1$) the linear regressions ($H_{1/3}/H_s = 1.00$ and $H_{max}/H_s = 1.81$) are in good agreement with the theoretical predictions that Longuet-Higgins (1952) derived for a narrowbanded spectrum (Table 3).

linear fit. The coefficient 1.61 was lower than assuming the Rayleigh distribution of Longuet-Higgins (1952), but was in good agreement with the prediction of Forristall (1978) and Casas-Prat and Holthuijsen (2010) (Table 3). The maximum crest height at the GoF were well predicted by Casas-Prat and Holthuijsen (2010), but the wave troughs ($\eta_{min}$) agreed better with Longuet-Higgins (1980).

The linear regression to the Suomenlinna data was $H_{max} = 1.58 H_s$ (Fig. 7 a). The ratio was lower compared to the GoF even though we would expect the higher $N$ at Suomenlinna (caused by shorter waves) to result in a higher single wave $H_{max}/H_s$. The disagreement with previous studies was also more pronounced (Table 3). We determined a linear fit with the spectral narrowness to be $H_{max}/H_s = 1.54 + 0.27\kappa^2$ (Fig. 6 b). This regression results in $H_{max}/H_s = 1.81$ for an theoretical infinitely peaked spectrum ($\kappa^2 = 1$), which is in good agreement with the theoretical derivations of Longuet-Higgins (1952)

that assumed a narrowbanded spectrum (Table 3). Nevertheless, the very low correlation between the variables ($r$=0.15) limits the confidence in this specific result. Vandever et al. (2008) found no connection between $H_{max}/H_{1/3}$ and $\nu$. The correlation between $H_{max}/H_{1/3}$ and $\kappa^2$ was zero also in our data, most probably because of the self scaling nature of $H_{max}/H_{1/3}$.

The maximum crest height, $\eta_{max}/H_s$, at Suomenlinna was only slightly lower than at the GoF (Table 3). If symmetry would be assumed, the maximum crest heights would be in perfect agreement with the estimates from the Rayleigh distribution of

Longuet-Higgins (1952), which was also found by Casas-Prat and Holthuijsen (2010). The troughs were slightly shallower in our data compared to e.g. Casas-Prat and Holthuijsen (2010), but were well described by the scaled Rayleigh distribution of Longuet-Higgins (1980). There was no correlation between $\eta_{max}/H_s$ (or $\eta_{min}/H_s$) and the spectral narrowness $\kappa^2$ ($r$=0.0).

None of the aforementioned dimensionless wave/crest heights had any correlation with the dimensionless depth, $k_p h$ ($r$=0.0). Together these results suggests that the main factor controlling the magnitude of the highest single waves was the spectral shape.

Thus, the differences to previous results were mainly caused by the violation of the assumption of a narrowbanded spectrum,





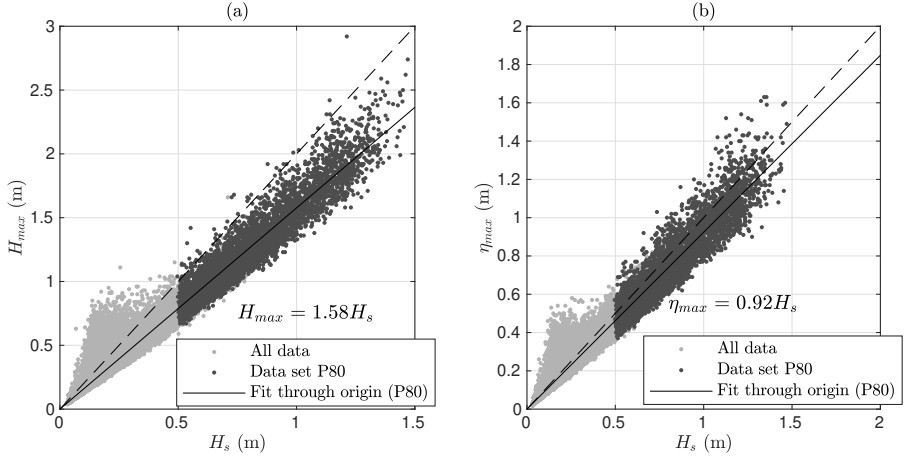

**Figure 7.** The wave height of the single highest wave ($H_{max}$, a) and the maximum crest height ($\eta_{max}$, b) relative to the significant wave height, $H_s$, at Suomenlinna.

not the assumption of deep water. The exceptions were the crest and trough heights, which exhibited no connection to the spectral width.

The maximum single wave measured at Suomenlinna was $H_{max}$=2.92 m ($H_{max}/H_s$=2.41), the maximum crest-height was $\eta_{max}$=1.54 m. This wave was measured during south-easterly winds ($U_d$=152 deg). It is evident from Fig. 6 (b) that a ratio

5   over 2 was not a rare occurrence, since it happened 45 times during the three year deployment period of the buoy. Still, the criteria of (roughly) $H_{max}/H_s$>2 is often taken as a definition for "rogue waves" (e.g. Onorato et al., 2002). Also Casas-Prat and Holthuijsen (2010) found thousands of single waves exceeding twice the significant wave height. The generation of rogue waves have been proposed to be controlled by modulation instability (Janssen, 2003), which is quantified using the Benjamin-Feir Index (Eq. (13)). Nevertheless, the correlation between BFI and $H_{max}/H_s$ (or $\eta_{max}/H_s$) was only 0.1 for the Suomenlinna

10   P80 data set (not shown). The lack of descriptive power of the BFI is expected, because the modulation instability is strongest for narrowbanded spectra—the exact opposite of the conditions that we have observed in the archipelago.

### 4.4   The zero-crossing period, $T_z$

As the significant wave height, the zero-crossing period, $T_z$, is one of the oldest wave parameters. Based on theoretical arguments about the Gaussian distribution of the water level displacement it can be calculated from the spectral moments as $T_{m_{02}}$

15   (Eq. (3)). Since this connection is based on theoretical assumption, it might not be valid for atypical spectral shapes, as the ones found in the archipelago.

We compared these two definitions of the zero-crossing period using a linear fit to the P80 data sets. For the GoF data the two definitions agreed well, with a linear fit giving a proportionality coefficient of 1.02. For the Suomenlinna data the linear fit was $T_z = 1.04 T_{m_{02}}$ (Fig. 5 c), meaning that the traditional definition of the zero-crossing period was quite robust and coincided





**Table 3.** The different wave height and crest height parameters at Gulf of Finland and Suomenlinna. The values have been determined using a linear fit through the origin of the P80 data set. The ratio $H_{1/3}/H_s$ was takes as reported in the litterature. The single wave statistics for the distributions given in the litterature have been determined using the individual number of waves for each wave record and the mean value of the spectral width parameter $\nu$ (when given).

| | $H_{1/3}/H_s$ | $H_{max}/H_s$ | $\eta_{max}/H_s$ | $\eta_{min}/H_s$ |
|---|---|---|---|---|
| **Gulf of Finland (GoF)** | | | | |
| Measured (this study) | 0.927 | 1.61 | 0.93 | -0.85 |
| Longuet-Higgins (1952)[a] | 1 | 1.80 | 0.90 | -0.90 |
| Forristall (1978)[b] | 0.942 | 1.64 | - | - |
| Longuet-Higgins (1980)[c] | 0.925 | 1.67 | 0.84 | -0.84 |
| Longuet-Higgins (1980)[c] ($\nu = 0.361$) | 0.951 | 1.72 | 0.86 | -0.86 |
| Casas-Prat and Holthuijsen (2010)[d] | 0.957 | 1.63 | 0.93 | -0.90 |
| **Suomenlinna (T2)** | | | | |
| Measured (this study) | 0.881 | 1.58 | 0.92 | -0.83 |
| Longuet-Higgins (1952)[a] | 1 | 1.84 | 0.92 | -0.92 |
| Forristall (1978)[b] | 0.942 | 1.68 | - | - |
| Longuet-Higgins (1980)[c] | 0.925 | 1.71 | 0.85 | -0.85 |
| Longuet-Higgins (1980)[c] ($\nu = 0.335$) | 0.958 | 1.77 | 0.88 | -0.88 |
| Casas-Prat and Holthuijsen (2010)[d] | 0.957 | 1.67 | 0.95 | -0.92 |

[a] Assuming a narrow banded spectrum in deep water, Gaussian water level elevations with respect to the still water level, and a Rayleigh distribution for the heights of single waves.

[b] Empirical Weibull fit to storm data.

[c] Empirical Rayleigh fit to the storm data of Forristall (1978).

[d] Empirical Rayleigh fit based on 15 years of measurements from four wave buoys.

well with the one calculated from the spectral moments over a wide range of spectral shapes. In the Suomenlinna data the ratio $T_z/T_{m_{02}}$ was only weakly correlated with the $\kappa^2$ narrowness parameter ($r = -0.18$). A linear fit ($T_z/T_{m_{02}} = 1.05 - 0.04\kappa^2$) still gave almost unity for a theoretical narrowbanded spectrum ($\kappa^2 = 1$).

The ratio $T_z/T_{m_{02}}$ at Suomenlinna was also correlated with the dimensionless depth $k_p h$ ($r = -0.16$). The explanation for
5  this, possible spurious, correlation might be the same as for the significant wave height, namely that more sheltering results in both a wider spectrum and a deeper dimensionless depth. Nonetheless, the sign of the correlation was what would be expected if the variations were really explained by the variations in dimensionless depth. The reason for the slight 4 % bias at Suomenlinna was therefore left undetermined.




## 4.5  Finding a characteristic frequency, $f_c$

Often the full spectrum is not available, and the characteristics of the wave field is described using a limited set of integrated parameters. If directionality is ignored, the choice is usually a measure for the height and a measure for the length, or equally well, the frequency. A unimodal spectrum, for example, is quite well described by the significant wave height and the peak

frequency. Nevertheless, the flat spectral shape in the archipelago leads to a low stability of the peak frequency. The mean frequency, again, is stable, but biased compared to the the peak frequency for the more unimodal spectra in the outer archipelago.

Young (1995) proposed a definition for the peak frequency, $f_p^q$, that is based on a weighted mean integral of the spectrum (Eq. (6)). This expression has a free parameter, $q$, that needs to be determined. We set out to study if any exponent of $q$ could produce a "characteristic" frequency (henceforth, $f_c$) that would be more stable than simply taking the argument maximum of

the spectrum, but still wouldn't be as biased as $f_m$ compared to the peak frequency. The challenge in choosing a value for $q$ is that minimizing the scatter suggest a low values for $q$ (with $q$=1 resulting in $f_m$), while minimizing the bias compared to $f_p$ requires a high value for $q$. To determine a "best estimate", we defined an error function:

$$\mathrm{Er}(q) = \left[ \langle f_p - f_p^q \rangle^2 + \left( \sigma(f_p^q) \right)^2 \right]^{\frac{1}{2}}, \tag{20}$$

where $\langle \cdot \rangle$ denotes the mean and $\sigma$ is the standard deviation. In other words, it's the norm of the bias and the standard deviation

of $f_p^q$.

We determined this error function for each location separately using he P80 data set. The minimum was achieved between $q = 3$ and $q = 5$, with the exception of the Sheltered archipelago sites ($q$ =2–3). These values are in line with $q = 4$ of Young (1995), but lower than $q = 8$ of Sobey and Young (1986), that the authors recommended for an alternative definition of the peak frequency.

In addition to a best estimate of $q = 4$ we compared the peak frequency to $f_p^q$ using the values $q = 1$ and $q = 10$. As a metric quantifying how different candidates for $f_c$ characterizes the spectrum, we determined the relative amount of energy that is carried by waves below the characteristic frequency, that is:

$$E_0(f_c) = \frac{1}{m_0} \left( \int_0^{f_c} E(f) \, df \right). \tag{21}$$

In the GoF data roughly 65 % of the energy was below the mean frequency regardless to the wind direction ($E_0(f_m) \approx 0.65$,

Fig. 8 c). For the peak frequency the respective value was 29 %, but it varied with the wind direction, being as high as 40 % for southerly winds. The southerly wind sector produces waves that are unaffected by the narrow fetch geometry of the gulf. They should therefore most closely resemble classic fetch limited spectra, although they might still be affected by swell propagating along the gulf, especially from the Baltic Proper in the west. With a choice of $q = 4$ the integrated parameter $f_p^q$ agreed well with the peak frequency for southerly winds in the GoF data set (Fig 8 c). For other wind directions, where the narrow fetch

effects came into play, $f_p^{q=4}$ resulted in slightly higher frequency estimates compared to $f_p$ (Fig. 8 c). Since the most dominant

**Figure 8.** The characteristic wave frequency $f_c = f_p^q$ compared to $f_p$ for different values of $q$ at the open sea (GoF, left column) and in the middle of the archipelago (Suomenlinna, right column). Note, that $f_m$ is identical to $f_p^{q=1}$.

wind directions are along the axis of the Gulf, it is clear that $f_p^{q=4}$ doesn't produce an unbiased estimate in a general sense. A choice of $q = 10$ introduced practically no bias, and can therefore be used as an alternative definition for the peak frequency (Table 4, Fig. 8 c).





For Suomenlinna $f_p^{q=4}$ also showed a good general agreement with the peak frequency, and the mean value of $E_0(f_p^{q=4})$ was almost identical (35 % vs. 34 %) to the one determined for the GoF (Fig. 8 b & d). The energy below the mean frequency was, on average, only 60 %, but this value depended strongly on the wind direction. For the southerly winds—where the spectral shape was most peaked—$E_0(f_m)$ agreed with the GoF data. For the wider spectra of the other wind directions the two sites

disagreed; especially for eastern winds the amount of energy below the mean frequency was only slightly above 50 %, which would be the value for a theoretical white noise spectrum. Also $E_0(f_p^q)$ varied with the wind direction for both $q = 4$ and $q = 10$. Even though a similar variation was seen also in the GoF, the easterly wind directions at Suomenlinna produced wave spectra where, on average, only 20 % of the energy was below the peak frequency—a situation that was not possible at the GoF.

Choosing $q = 4$ resulted in in $E_0(f_c)$ being roughly between 30 % and 40 % at both in the open sea (GoF) and in the archipelago (Suomenlinna). While the integrated definition using $q = 4$ was not identical to the peak frequency, it had the advantage of describing a similar characteristic feature in both locations. Namely, in a mean sense, 30–40 % of the wave energy was contained by waves with a frequency lower than $f_c$. A consistency in this respect is important when studying e.g. erosion, or the waves' effect on floating objects. Using $q = 10$ is attractive as its bias with respect to the peak frequency was

small at all locations (Table 4). On the other hand, the scatter (as measured by the standard deviation) was only reduced slightly compared to the peak frequency.

## 5 Discussion

The peak frequency is a very noisy parameter for the flat spectral shape found in the archipelago. This noise is connected to the sampling variability of a spectrum that has a wide frequency range with an almost constant variance density—the peak

frequency is determined by the variability of the "peaks" in this region. Using the integrated definition $f_p^q$ of Eq. (6) with $q = 10$ reduced the scatter slightly while adding no bias compared to the peak frequency (Table 4). This can therefore be recommended as an alternative definition for the peak frequency in archipelago conditions.

The mean frequency is well defined and stable, and is therefore a good choice to describe certain aspects of the wave field. It is still important to realize that, while the mean frequency is well defined both in the open sea and in the archipelago, it doesn't

necessarily characterize the wave fields in a similar fashion. The amount of energy contained in the wave spectrum below the mean frequency was determined by the shape of the spectrum: for the GoF it was roughly 65 %, while in the archipelago at Suomenlinna it was lower, depending on the wind direction (Fig. 8). The energy below the peak frequency also varied at both locations, being as low as 20 % at Suomenlinna for the wind sector $110 \leq U_d \leq 130$. These differences could introduce large uncertainties, especially if wave-bottom interactions or the effect on floating objects are quantified using a single frequency.

Equation (6) with $q = 4$ seemed to be a balanced choice for a characteristic frequency, $f_c$, for a wide range of conditions. First, the energy of waves longer than those with a frequency $f_c$ was between 30 % and 40 % both in the open sea and in the archipelago. Second, it was more stable than the peak frequency, while still reflecting the energy maximum of the spectrum more closely than the mean frequency. Last, for waves growing from a straight shoreline (southerly winds in the GoF) it became





**Table 4.** The mean values and the scatter (standard deviation) of the characteristic wave frequency $f_c := f_p^q$ for three different values of $q$ (see Eq. 6) compared to that of the peak frequency, $f_p$. Note, that $f_p^q = f_m$ for $q = 1$.

| Name (Code) | All data $\langle f_c \rangle / \langle f_p \rangle$ | | | All data $\sigma(f_c)/\sigma(f_p)$ | | | Data set P80 $\langle f_c \rangle / \langle f_p \rangle$ | | | Data set P80 $\sigma(f_c)/\sigma(f_p)$ | | |
|---|---|---|---|---|---|---|---|---|---|---|---|---|
| | $q=10$ | $q=4$ | $q=1$ | $q=10$ | $q=4$ | $q=1$ | $q=10$ | $q=4$ | $q=1$ | $q=10$ | $q=4$ | $q=1$ |
| Gulf of Finland (GoF) | 1.01 | 1.04 | 1.23 | 0.96 | 0.92 | 0.73 | 1.00 | 1.02 | 1.26 | 0.95 | 0.87 | 0.79 |
| **Outer archipelago** | | | | | | | | | | | | |
| Harmaja (O1) | 1.00 | 1.02 | 1.19 | 0.94 | 0.85 | 0.52 | 1.00 | 1.01 | 1.18 | 0.82 | 0.68 | 0.54 |
| Isosaari (O2) | 1.00 | 1.03 | 1.18 | 0.94 | 0.88 | 0.60 | 1.00 | 1.02 | 1.21 | 0.94 | 0.90 | 0.85 |
| Berggrund (O3) | 1.01 | 1.04 | 1.19 | 0.95 | 0.86 | 0.58 | 1.01 | 1.03 | 1.25 | 0.90 | 0.87 | 0.91 |
| **Transition Zone** | | | | | | | | | | | | |
| Itä-Villinki (T3) | 1.01 | 1.05 | 1.27 | 0.94 | 0.88 | 0.57 | 1.02 | 1.09 | 1.33 | 0.87 | 0.76 | 0.57 |
| Lansikari (T1) | 1.00 | 1.03 | 1.23 | 0.92 | 0.84 | 0.57 | 1.00 | 1.03 | 1.28 | 0.87 | 0.79 | 0.78 |
| Suomenlinna (T2) | 1.01 | 1.04 | 1.16 | 0.93 | 0.83 | 0.45 | 1.01 | 1.04 | 1.26 | 0.90 | 0.85 | 0.72 |
| **Inner archipelago** | | | | | | | | | | | | |
| Jätkäsaari (I3) | 1.01 | 1.05 | 1.16 | 0.90 | 0.77 | 0.38 | 1.01 | 1.09 | 1.24 | 0.80 | 0.59 | 0.27 |
| Hernesaari (I1) | 1.01 | 1.05 | 1.18 | 0.90 | 0.79 | 0.48 | 1.01 | 1.05 | 1.21 | 0.85 | 0.73 | 0.44 |
| Ruumiskari (I2) | 1.01 | 1.04 | 1.15 | 0.90 | 0.75 | 0.36 | 1.04 | 1.15 | 1.48 | 0.93 | 0.81 | 0.35 |
| **Sheltered archipelago** | | | | | | | | | | | | |
| Koivusaari (S1) | 1.00 | 1.00 | 0.97 | 0.89 | 0.73 | 0.35 | 1.00 | 1.00 | 0.97 | 0.83 | 0.58 | 0.27 |
| Ramsinniemi (S2) | 1.00 | 0.99 | 0.94 | 0.92 | 0.75 | 0.36 | 0.99 | 0.98 | 0.89 | 0.85 | 0.71 | 0.38 |
| Vuosaaren satama (S3) | 1.00 | 0.98 | 0.91 | 0.84 | 0.64 | 0.32 | 0.99 | 0.98 | 0.91 | 0.85 | 0.68 | 0.29 |
| Talosaari (S4) | 1.00 | 0.99 | 0.90 | 0.88 | 0.68 | 0.35 | 0.99 | 0.98 | 0.87 | 0.91 | 0.77 | 0.38 |

a non-biased estimate of $f_p$. Young (1995) recommended Eq. (6) with $q = 4$ as an alternative definition for the peak frequency. Our results still indicate that this definition is a biased estimated for the peak frequency under more complex conditions, such as in the archipelago, or for waves affected by the narrow fetch geometry of the Gulf of Finland. We therefore recommend that it should not used as an alternative definition of the peak frequency in archipelago conditions, but it can be added as a new parameter with a distinct name, e.g. the characteristic frequency. Nonetheless, all aspects of the complex archipelago wave field cannot be described by a single frequency. A proper description would require adding at least one more parameter quantifying





the width of the "energy carrying" range of the spectrum. A promising candidate for a parameterization of this kind is the $\kappa^2$ parameter by Battjes and van Vledder (1984) or the Goda peakedness parameter (Goda, 1970).

    We also found that the d.o.f. of the wave variance ($m_0$) closely reflected the spectral width, and they seemed to correlate well with the narrowness parameter $\kappa^2$ (Fig. 4). Wider ("flatter") spectra had higher degrees of freedom, which lead to
smaller confidence limits for the measured significant wave height (Table 2). It follows that, when evaluating a wave model in archipelago conditions, a constant performance will lead to a smaller scatter index (or normalized root-mean-square-error) inside the archipelago compared to the open sea. A theoretical "perfect model" would therefore show an increased accuracy for the significant wave height when moving through the archipelago, while conversely showing a decrease in the accuracy in the peak frequency. We do, however, want to point out that the latter effect is much stronger.

The spectral shape affected the relation between the different definitions of the significant wave height ($H_{1/3}$ vs. $H_s$). The ratio $H_{1/3}/H_s$ varied, in a mean sense, as a function of the spectral narrowness $\kappa^2$ (Fig. 6 a). Regardless of the scatter, this connection suggested a decreased height of the highest single waves compared to the total variance for a wider spectrum. The highest single wave $H_{max}/H_s$ was, indeed, statistically lower at Suomenlinna compared to the open sea (Table 3). A connection to $\kappa^2$ was also found (Fig. 6 b), although with a very weak correlation ($r=0.15$). The low correlation between the
highest single wave and the spectral width might partly be explained by the higher number of waves associated with a wider spectrum. If the average values at Suomenlinna are viewed as a function of the wind direction (as d.o.f.($m_0$) in Fig. 5 a), the relevant parameters have a variation of $494 \leq N \leq 577$ and $1.54 \leq H_{max}/H_s \leq 1.60$. Assuming a single Rayleigh distribution, the variation in $N$ would cause a difference of 1.1 % in estimates for $H_{max}/H_s$, while the observed variation in the average value of $H_{max}/H_s$ was 3.5 %.

The reduction of the single highest wave in a wider spectrum has been explained by the de-correlation of the following crests and troughs: a deep trough is less likely to be followed by a high wave crest, even if the maximum and minimum water levels are not affected (e.g. Tayfun, 1983). This is also supported by our data, since we found no connection between the crests (or troughs) and the spectral width. Goda (1970) found that in computer-simulated data the height of the single waves followed a Rayleigh distribution regardless of the spectral width (as quantified by $\varepsilon$ of Cartwright and Longuet-Higgins
(1956)). Nevertheless, based on a very extensive data set, Casas-Prat and Holthuijsen (2010) pointed out that the use of other distributions can have an advantage over the Rayleigh distribution for large number of $N$. Further research is needed to resolve the open questions regarding the affect of spectral width on the distribution of the height of single waves.

    We found the variations in spectral shape, not the finite water depth at the measurement site, to explain the deviations from previous deep water results. Still, the finite depth—together with the sheltering effect of the islands—plays an important role
in shaping the wave field through bottom friction and depth induced wave breaking before the waves arrive to a certain location in the archipelago. The effect of the finite depth further away from the measurement location was thus captured by the spectral shape, even though e.g. shoaling effects caused by the local water depth did not explain our results.





## 6 Conclusions

An extensive field measurement campaign consisting of wave buoy measurements from 14 locations was performed in the Helsinki archipelago during 2012–2018. Multi-year time series were available from two operational wave buoys in the middle of the Gulf of Finland (GoF) and in the middle of the archipelago (Suomenlinna). Measurements from the other sites in the archipelago lasted for about a month. These measurements were used to study the shape of the wave spectrum in the archipelago and the consequences the variations of the spectrum have for derived wave parameters.

The mean spectral shape in the middle of the GoF was unimodal with a distinct peak. No peak was identifiable close to the shoreline, where the spectrum was best described by a wide energy carrying range with almost constant variance density. At Suomenlinna, located in between these two extremes, the spectral shape varied strongly with the wind direction because of the anisotropic fetch conditions. For south-westerly, and especially southerly, winds the spectral shape was peaked. For easterly winds the spectral shape was wide, being close to that of the sites near the shore.

The mean shape of the spectra were well quantified by the spectral narrowness parameter ($\kappa^2$) of Battjes and van Vledder (1984), but a scatter still persisted. The width parameter $\nu$ of Longuet-Higgins (1975) had no predictive value, possibly because of the challenges imposed by measuring the short waves in the archipelago with wave buoys. The spectral width was also connected to the degrees of freedom of the wave variance ($m_0$), with a wider spectrum having higher degrees of freedom. As a direct consequence, the confidence limits for the measured significant wave height are lower inside the archipelago compared to the open sea (Table 2).

The spectral shape affected the ratio $H_{1/3}/H_s$, with a wider spectrum resulting in a lower ratio ($H_s$ was defined using the variance). The ratio between the two definitions of significant wave height was determined to be $H_{1/3} = 0.881 H_s$ at Suomenlinna, but the ratio varied as a function of the spectral narrowness $\kappa^2$ (Fig. 6 a). The effect of the spectral shape on the ratio $T_z/T_{m_{02}} = 1.04$ was weak.

The highest single wave $H_{max}/H_s$ was, on average, 1.58 at Suomenlinna, which is lower both compared to the open sea measurements at GoF (1.61) and to estimates using the literature (1.67–1.84, Table 3). Our results suggests that the deviation in $H_{max}/H_s$ from previous studies are mainly caused by a wider spectral shape (Fig. 6 b), not by the finite water depth. Nevertheless, the weak correlation found in the data can offer no solid conclusions, and the issue warrants further research.

The traditional peak frequency, $f_p$, was practically undefinable in the archipelago. As a compromise between scatter and bias with respect to $f_p$, the integrated frequency weighted by $E(f)^4$ was proposed as a "characteristic" frequency, $f_c$. This definition was applicable over a wide range of wave conditions, and functioned as a non-biased estimated for $f_p$ in waves growing from a straight shoreline. Nevertheless, is cannot be used as an alternative definition for the peak frequency in more complex conditions inside the archipelago, or even in the open sea if the wave growth is affected by narrow fetch geometry.

## 7 Data availability

The measurement data will be made available in a public repository upon the publication of the final paper.





*Author contributions.* JVB initiated the study based on previous conceptualizations of KK and HP. KK and JVB took part in designing the field measurements. JVB produced the methodology and performed the analysis. JVB did the visualization. JVB wrote the manuscript with contributions from HP and KK. HP supervised the study.

*Competing interests.* The authors declare that they have no conflict of interest.

5 *Acknowledgements.* We want to acknowledge the work done by the technical staff at FMI, namely Mr. Tuomo Roine, Ms. Heini Jalli, and Ms. Riikka Hietala. The efforts of Mr. Hannu Jokinen in processing the wave raw wave buoy data is also gratefully acknowledged. Most of the wave buoy observations in this study have been collected through work commissioned by the City of Helsinki. The data is used in this paper with their kind permission. This work has been partially funded by Arvid och Greta Olins Fond (Svenska kulturfonden, 17/103386) and the Doctoral school in natural sciences (University of Helsinki).



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
