# Peer review of "The wave spectrum in archipelagos"

_Ocean Science, 2019_

## Referee Comment (RC1) · Anonymous Referee #1 · 1 Aug 2019

Review of "The wave spectrum in archipelagos" by Jan-Victor Björkqvist, Heidi Pettersson, and Kimmo K. Kahma, manuscript os-2019-59

General Comments

This is an excellent paper, adding new observational insights into wave fields in archipelagos, adding substantial new insight to an already large volume of wave observation literature. Well done!

Specific Comment / questions

Page 5, section 2.3.1: In computing wave parameters from the spectrum, do the authors use a parametric tail, or is the integration stopped at the highest frequency of the observations. See also Page 3, line 16.

Page 7, line 13: Very small wave heights are not used, but in Table 1 mean wave

heights in the sheltered locations is still as small as 0.05m. Can such small mean wave heights in the sheltered areas be trusted?

Page 10, lines 15-18: How did the authors concluded that peaks in the spectrum were due to refracted wave components?

Page 23, Section 5: I enjoyed the discussion of defining a representative frequency. It seems to me however, that the focus is mostly on getting a table parameter. Coming from an operational / application side, the usage of the parameter should be considered too (erosion, loads, roughness, etc.).

Page 26, Section 6: The data selection results in wind sea presence in all cases. This focu on wind seas needs to be repeated in the Conclusions.

Technical Correction

Page 3, line 8: Add text in red "While observations at 12 out of 14 . . ."

Page 3, line 29: ". . .time series, used to compute a single wave spectrum."

Page 7, line 16; please confirm that the 80th percentile is relative to the entire time series at the given location.

Page 14, Table 4: Labeling of panels is not consistent. Why is (a) labeled d.o.f, whereas all other panels are d.o.f. too?

---

## Referee Comment (RC2) · Anonymous Referee #2 · 8 Aug 2019

General comments: The authors have analysed a large number of wave observations from a complex shoreline structure, the Finnish archipelago. The paper has many interesting results worth while to publish, like the relation between H1/3 to Hs, the discussion on critical frequency versus peak frequency, how the shape of the spectrum flattens when going more inside the islands. Several interesting parameters are used in the analyse, i.e. the critical frequency from Young (1995), the spectral narrowness parameter $\kappa^2$ from Battjes and van Vledders (1984), the degrees of freedom (d.o.f.) from Donelan and Pierson (1983), and the paper has some good results using these, certainly worth while to publish. But the paper is difficult to read. It needs a considerable rebuilding. Motivation is only vaguely mentioned in between a number of references to different papers showing the authors have done a good research in this analysis. The database seems to have important weaknesses. The data in the inner and sheltered zones have average Hs of around 20 cm decreasing to 2-5 cm. And it seems they are

measured with buoys with 40 cm diameter. With sampling frequency 1.28 Hz. And the measuring period is only 14-31 days for these. Can such measurements be at all reliable? The authors do deal in what is a second part of the paper with mostly with T2 versus GoF, where measuring period is 2-3 years. But this should be more clearly stated in the paper.

Specific comments: The paper should be rewritten for an easier access of results to community. Only a few comments are given here.

Regarding motivation: It is believed that questions to answer is how much of offshore wave energy enters through the islands, and in what form (distribution in frequency, spectral shape...). Reduction factors are mentioned, without saying how many cases are involved. How is low frequency energy reduced inside the archipelago?

The paper is difficult to read for several reasons. Site description: an overview map is needed for the understanding of fetches. I would suggest one with only land contours, and perhaps the 40m and 80m isolines would help, covering the area of importance for the GOF and T2 point. The map in figure 1 is difficult to 'read' because land has a colour difficult to identify in between strong variations in depth.

The overview of the database in Table 1 comes too late. Names of stations are given in the text here and there, the identifications ('T1', 'T2') would help to be given together with the names.

Database: What conditions do we have in general at the two sites with wind measurements, a) for the 2-3 year period in last part of paper, and in the periods where the inner sites are included.

Technical corrections No details at this point.

---

## Author Comment (AC1) · 12 Sep 2019

We would like to thank the reviewers and the editorial staff for the swift and timely handling of our submission.

Anonymous Referee 1

**R1: General Comments** This is an excellent paper, adding new observational insights into wave fields in archipelagos, adding substantial new insight to an already large volume of wave observation literature. Well done!

***Our response****: We are pleased that you liked our manuscript. Thank you for taking the time to review it.*

**R1: Specific Comment / questions**

[Figure]

**R1**: Page 5, section 2.3.1: In computing wave parameters from the spectrum, do the authors use a parametric tail, or is the integration stopped at the highest frequency of the observations. See also Page 3, line 16.

***Our response**: We chose not to use a parametric tail in this study. The reason is that the material we present on the wave spectrum in the archipelago are new, and we want our results to reflect only the measurements, not any theoretical assumptions we have made about the tail of the spectrum. That said, an $f^{-4}$ parametric tail would be very reasonable if such an approach was taken. We feel that it is better to leave the uncertainty that stems from the (at times) poorly measured tails in our results. Nonetheless, this limitation should be stated more clearly (please see our response to your next comment).*

*We have edited the manuscript to clearly state that no parametric tail was used (Page 3, line 23-24 and page 7, line 6)*

**R1**: Page 7, line 13: Very small wave heights are not used, but in Table 1 mean wave heights in the sheltered locations is still as small as 0.05m. Can such small mean wave heights in the sheltered areas be trusted?

***Our response**: This is a fair point. We think that our results show that they cannot be completely trusted, which is seen in that the attempts to quantify the spectral narrowness $\kappa^2$ and the degrees of freedom in $m_0$ gives conflicting results with respect to the other locations (Table 2). We still wanted to include them for two reasons: i) we think they add value in the (partially qualitative) analysis in Figure 2, and ii) we think it is also important to present the limitations and challenges in measuring the wave field inside the archipelago using standard instrumentation.*

*Referee 2 also commented on this issue. Please also see our response to that comment.*

*We have edited the manuscript to more clearly state the limitations of the measure-*

ments at locations S1-S4 (page 12, line 13; page 13, lines 20-24 and page 27 lines 10-18).

**R1**: Page 10, lines 15-18: How did the authors concluded that peaks in the spectrum were due to refracted wave components?

***Our response***: *A directional spectrum from Jätkäsaari (I3) has been presented in a conference paper that studied the reflection of waves from a steep shore (Björkqvist et al., 2017). There the directional separation of the two wave systems are clearly visible. The narrowness (in frequency) of the wave component is because wave slightly shorter (or longer) are refracted in a different manner and therefore propagate to a slightly different location.*

*We have added a reference to the above-mentioned conference paper on page 11, lines 22-24 of our manuscript.*

**R1**: Page 23, Section 5: I enjoyed the discussion of defining a representative frequency. It seems to me however, that the focus is mostly on getting a table parameter. Coming from an operational / application side, the usage of the parameter should be considered too (erosion, loads, roughness, etc.).

***Our response***: *We regret that the fundamental nature of the study might leave the practical considerations slightly lacking. We tried to tie the properties to practical issues in Section 4, but the nature of the characteristic frequency is admittedly a bit light. While we think that the role of the characteristic frequency will be central, even if it in itself cannot function as the sole quantifier of an archipelago wave field.*

*To illustrate our point we calculated the root-mean-square near bottom velocities ($U_{rms}$) from the spectra at Suomenlinna (T2), and these values were compared to a simplified monochromatic approach (see the Figure below). The comparison shows the comparison for both the peak frequency and the new characteristic frequency. While we could argue that the new frequency reduces the scatter somewhat, the truth is that*

*both approaches are fundamentally wrong in an archipelago setting, and goes against the core idea of this paper.*

*The main point is that the characteristic frequency defines where the energy is concentrated in some absolute sense (in Hz), while the spectral width defines how widely the energy is spread around this particular frequency. Using only one or the other can clearly not capture e.g. wave-bottom interactions. More elaborate parametrizations exist, but they can depend on some assumed spectral shape. The correct approach to build a parameterization for the archipelago would therefore require us to find an analytical functional form that uses the total energy (or $H_s$), the characteristic frequency, and some kind of spectral width. One reasonable parameterization would then be to approximate the spectrum with a box covering the dominant frequency interval of the spectrum, and deduce some expression for $U_{rms}$ from that.*

*Many practical applications of the characteristic frequency therefore depend on finding a good functional form for an archipelago type spectrum. This is to a high degree not trivial. We are also reluctant to present any results that we know to be, in a theoretical sense, flawed.*

*What we can do is to make it clear in the discussion (Section 5.2) that from a practical point of view the results about the characteristic frequency is still a stepping stone, and outline how the results in this paper provides some tools for further, both theoretical and practical, research. We have also edited the end of the abstract and the conclusions to this affect.*

**R1**: Page 26, Section 6: The data selection results in wind sea presence in all cases. This focu on wind seas needs to be repeated in the Conclusions.

***Our response****: This has been added to the conclusions. (Page 28, lines 3-5)*

**R1: Technical Correction**

**R1**: Page 3, line 8: Add text in red "While observations at 12 out of 14 . . ."

*Our response: Thank you for catching this. The sentence has been corrected in the manuscript.*

**R1**: Page 3, line 29: ". . .time series, used to compute a single wave spectrum."

*Our response: This sentence has been corrected in the manuscript.*

**R1**: Page 7, line 16; please confirm that the 80th percentile is relative to the entire time series at the given location.

*Our response: We have modified the manuscript to state that the 80th percentile is relative to all the available data for the archipelago measurements (page 9 lines 17-18). For the GoF the 80th percentile is calculated from the years 2016-2018 to be comparable to the Suomenlinna (T2) measurements.*

**R1**: Page 14, Table 4: Labeling of panels is not consistent. Why is (a) labeled d.o.f, whereas all other panels are d.o.f. too?

*Our response: You are right. We have changed the label in panel (a) in Figure 4. It now reads "Wind direction $U_d$"*

**Other changes in the manuscript:**

1) *Page 7: We added a subscript "BFI" to the variables $\alpha$ and $\beta$ in Eq. (13), since the variable $\beta$ is used in Eq. (14) in a difference context. The variable $\alpha$ is used in another context in Eq. (19).*

2) *Page 8: We have changed the normalization of the spectra slightly by defining the coefficient $\beta$ as a mean over $E(f)f^4$ instead of a mean over only $E(f)$. This has no real consequences for the results (which can be seen from the redrawn Figures 2 and 3), but this approach is slightly more flexible. From a theoretical point of view the averaging can now be done over different frequency intervals for each spectrum, if we know that a $f^{-4}$ tail exists. While we use a fixed frequency in this paper, we wanted to introduce this slightly more general method since it might turn out to be useful in later studies,*

*and our upcoming results will then be more consistent with the current ones.*

3) *We added the information that two of the measurement sites were made for research purposes outside the commissioned work by the City of Helsinki. This has no consequences for the paper, but we wanted to represent the data more accurately (page 3 lines 12-14).*

4) *We corrected that the Harmaja wind station is 2 km from the Suomenlinna (T2) wave buoy (the Harmaja wave buoy was roughly 5 km from the T2 wave buoy, which is where the incorrect distance stemmed from). (page 5, line 3). This has no consequences for the results of the paper.*

5) *Some minor changes to the language and adding sentences explaining the reasoning. These are visible in the track changes version of the manuscript.*

**References:**

*Björkqvist, J.-V., Vähäaho, I., and Kahma, K. K.: Spectral field measurements of wave reflection at a steep shore with wave damping chambers, in: WIT Transactions on the Built Environment, vol. 170, pp. 185–191, doi:10.2495/CC170181, 2017.*

[Figure]

[Figure]

**Fig. 1.** Near-bottom velocities (Urms) calculated from spectrum compared to monochromatic approaches using fp (a) and fc (b).

---

## Author Comment (AC2) · 12 Sep 2019

We would like to thank the reviewers and the editorial staff for the swift and timely handling of our submission.

Anonymous Referee 2:

**R2: General Comments** The authors have analysed a large number of wave observations from a complex shoreline structure, the Finnish archipelago. The paper has many interesting results worth while to publish, like the relation between H1/3 to Hs, the discussion on critical frequency versus peak frequency, how the shape of the spectrum flattens when going more inside the islands. Several interesting parameters are used in the analyse, i.e. the critical frequency from Young (1995), the spectral narrowness parameter $\kappa^2$ from Battjes and van Vledders (1984), the degrees of freedom (d.o.f.) from Donelan and Pierson (1983), and the paper has some good results using these,

certainly worth while to publish.

*Our response*:*Thank you for putting in the effort to grasp our paper correctly even though it was lacking in clarity. It is greatly appreciated.*

**R2: General Comments (continued)** But the paper is difficult to read. It needs a considerable rebuilding. Motivation is only vaguely mentioned in between a number of references to different papers showing the authors have done a good research in this analysis. The database seems to have important weaknesses. The data in the inner and sheltered zones have average Hs of around 20 cm decreasing to 2-5 cm. And it seems they are measured with buoys with 40 cm diameter. With sampling frequency 1.28 Hz. And the measuring period is only 14-31 days for these. Can such measurements be at all reliable? The authors do deal in what is a second part of the paper with mostly with T2 versus GoF, where measuring period is 2-3 years. But this should be more clearly stated in the paper.

*Our response*: *We have rewritten certain parts of the manuscript for clarity. General motivation has been added to the introduction (page 2, lines 24-30). The approach using only the T2 data has been clarified on page 13, lines 25-30. The discussion part of the paper has been mostly rewritten to focus less on technical issues and instead do more to give the reader a possibility to grasp the meaning of our results.*

*The possible problems of the most sheltered locations (S1-S4) were also raised by Referee 1 (please also see our response to that comment). It is true that these measurements are not completely reliable, which can be seen by the mismatch to the other locations in the quantifying numbers in Table 2. This is one of the reasons we used the data from T2, as you pointed out. The short measurement periods are undoubtedly a limitation, but no better data set for archipelagos exist.*

*We still want to point out that there are two things that increase our trust in that the data is good enough to draw the conclusions we are drawing: i) the transition in Figure 2 presents a gradual and stable transformation of the mean spectral shape. An*

*instability in these averaged shapes would be a symptom of an insufficient amount of data. ii) the locations T1 and T2 are (geographically) close to each other, and their mean spectral shape are very similar even though T1 is the smallest data set and T2 is the largest. Also the quantifying numbers in Table 2 between T1 and T2 are quite consistent compared to more/less sheltered locations.*

*We have edited the manuscript to clearly present and discuss the limitations of our data set and how our analysis methods are chosen to minimize the problems (page 12, line 13; page 13, lines 20-30 and page 27 lines 10-18).*

**R2: Specific Comments** The paper should be rewritten for an easier access of results to community. Only a few comments are given here.

***Our response****: We have rewritten the manuscript to the best of our ability by taking into account the relevant comments you have presented. Especially the discussion part of the paper should now aid in giving a better understanding of our results.*

**R2**: Regarding motivation: It is believed that questions to answer is how much of offshore wave energy enters through the islands, and in what form (distribution in frequency, spectral shape...). Reduction factors are mentioned, without saying how many cases are involved. How is low frequency energy reduced inside the archipelago?

***Our response****: The question is not exactly how the low frequency energy is reduced, but how the propagated low frequency energy compares to the local fetch limited wave system. Even though the energy at the spectral peak in the GoF is heavily attenuated (even to practically 0 for the sheltered location), the relative amount of "long waves" to the local wave system is still larger in the sheltered areas. In other words, in the open sea areas there is no significant contribution from waves longer than the peak. In the archipelago there exists a significant amount of energy in waves that are longer than the local fetch-limited waves. An absolute attenuation goes hand in hand with a relative enhancement.*

*While attenuation factors for different fixed frequencies could be calculated, they would not be very informative for the main question of the spectral shape, since the local wave field is not constant throughout the archipelago. Our attempt to capture the relevant features of the spectral transformation is the narrowness parameter. A full description of the problem would require a parameterization of the archipelago wave spectrum, perhaps using the characteristic frequency, the total energy, and some spectral width parameter. This is, however, not a trivial question, which is why it was left unsolved in this paper. We hope to be able to completely resolve the issue of fitting an analytical parameterized function to our data in upcoming studies. This issue is discussed in Section 5.2 of the discussion.*

*The attenuation coefficients given in Section 2.5 were not meant to be used for any deeper analysis, but were only used to check that our a priori guess of how the sites should be divided up was not completely unreasonable. The results of Fig. 2 and Table 2 further show that the grouping was consistent, and provides a much more elaborate quantification of the spectral evolution than simple attenuation coefficients.*

**R2**: The paper is difficult to read for several reasons. Site description: an overview map is needed for the understanding of fetches. I would suggest one with only land contours, and perhaps the 40m and 80m isolines would help, covering the area of importance for the GOF and T2 point. The map in figure 1 is difficult to 'read' because land has a colour difficult to identify in between strong variations in depth.

***Our response**:Thank you for your suggestion. We think the map in Figure 1 gives a good overview, so we are reluctant to remove it. We have, however, added a blow up of the T2 region with only a couple of isolines (Fig. 2). The land has also been changed to green so that it is not confused with the depth variations.*

**R2**: The overview of the database in Table 1 comes too late. Names of stations are given in the text here and there, the identifications ('T1', 'T2') would help to be given together with the names.

*Our response*:We have added the reference to Table 1 already in the first sentence of section 2.1 (page 3). We have also moved the place of the table to be earlier in the manuscript, but the exact layout will be decided in production. We have added the identification codes to the names where they were missing in the text.

**R2**: Database: What conditions do we have in general at the two sites with wind measure- ments, a) for the 2-3 year period in last part of paper, and in the periods where the inner sites are included.

*Our response*: This information has been provided in the manuscript (page 5 lines 8-9) and as two extra columns in Table 1.

**Other changes in the manuscript:**

1) *Page 7: We added a subscript "BFI" to the variables $\alpha$ and $\beta$ in Eq. (13), since the variable $\beta$ is used in Eq. (14) in a difference context. The variable $\alpha$ is used in another context in Eq. (19).*

2) *Page 8: We have changed the normalization of the spectra slightly by defining the coefficient $\beta$ as a mean over $E(f)f^4$ instead of a mean over only $E(f)$. This has no real consequences for the results (which can be seen from the redrawn Figures 2 and 3), but this approach is slightly more flexible. From a theoretical point of view the averaging can now be done over different frequency intervals for each spectrum, if we know that a $f^{-4}$ tail exists. While we use a fixed frequency in this paper, we wanted to introduce this slightly more general method since it might turn out to be useful in later studies, and our upcoming results will then be more consistent with the current ones.*

3) *We added the information that two of the measurement sites were made for research purposes outside the commissioned work by the City of Helsinki. This has no conse-quences for the paper, but we wanted to represent the data more accurately (page 3 lines 12-14).*

4) *We corrected that the Harmaja wind station is 2 km from the Suomenlinna (T2) wave*

*buoy (the Harmaja wave buoy was roughly 5 km from the T2 wave buoy, which is where the incorrect distance stemmed from). (page 5, line 3). This has no consequences for the results of the paper.*

5) *Some minor changes to the language and adding sentences explaining the reasoning. These are visible in the track changes version of the manuscript.*

---

## Editor Decision (ED1)

P2 line 31: 'properties need'

P7 line 15: 'calculation **of** the peak frequency'

P8 eqn (11): '$fm_{02}$' not '$fm20$'?

P15 line 12: 'have  more d.o.f.'

P19 line 2: 'is  often'

P20 line 20: 'differences from'

P21 line 6: 'criteria' should be 'criterion'

P21 line 8: 'have' should be 'has'

P23 line 16: 'he' should be 'the'

P27 line 21: 'similar order of magnitude to'

---

## Author Response (AR2)

Authors response to comments for "The wave spectrum in archipelagos" (os-2019-59)

The Topic editor provided us with the following comments:

P2 line 31: 'properties need'
Our response: This has been corrected

P7 line 15: 'calculation **of** the peak frequency'
Our response: This was corrected to "for calculating the peak frequency", since this was our original intent.

P8 eqn (11): 'fm 02 ' not 'fm20'?
Our response: This has been corrected in the equation, and also in the text.

P15 line 12: ' have  more d.o.f.'
Our response: This has been corrected

P19 line 2: 'is  often'
Our response: This has been corrected

P20 line 20: 'differences from'
Our response: This has been corrected

P21 line 6: 'criteria' should be 'criterion'
Our response: This has been corrected

P21 line 8: 'have' should be 'has'
Our response: This has been corrected

P23 line 16: 'he' should be 'the'
Our response: This has been corrected

P27 line 21: ' similar order of magnitude to'
Our response: This has been corrected

**Other corrections:**

1) We have corrected minor grammatical errors in the text, but nothing that affects the substance.

2) We corrected one number (from 3.5% to 3.9%) on Page 26, Line 7 (in the final manuscript). This was apparently a typo, but the change has no consequences for the paper.

3) We moved Table 3 to an earlier point in the manuscript, since it is referenced in the text before figures 7 and 8. No changes was made to the actual Table.

4) We converted the figures containing scatter plots to raster (png) images, since the vector pdf-version swere over 25 MB and we found no straightforward way to reduce the size otherwise. No changes were made to the actual figures.

5) We have published the data in an open repository. The DOI to the data set was added to the "Data availability" section of the manuscript.